# The Impact of Urbanization on the Supply–Demand Relationship of Ecosystem Services in the Yangtze River Middle Reaches Urban Agglomeration

Jie Gong [1,2,†], Xin Dai [3,4,†], Lunche Wang [3,4,*], Zigeng Niu [3,4], Qian Cao [3,4] and Chunbo Huang [3,4]

1   Institute of Geological Survey, China University of Geosciences, Wuhan 430074, China; gongj@cug.edu.cn
2   Wuhan Huaxin Lianchuang Technology Engineering Co., Ltd., Wuhan 430000, China
3   Hubei Key Laboratory of Regional Ecology and Environmental Change, School of Geography and Information Engineering, China University of Geosciences, Wuhan 430074, China; daixin5827@cug.edu.cn (X.D.); nzg@cug.edu.cn (Z.N.); caoqian@cug.edu.cn (Q.C.); huangchunbo@cug.edu.cn (C.H.)
4   Hubei Luojia Laboratory, Wuhan 430079, China
*   Correspondence: wang@cug.edu.cn; Tel.: +86-133-4988-9828
†   These authors contributed equally to this work.

**Abstract:** The urbanization process can alter the structure of urban land use and result in variations in urban ecosystem services (ESs). Researching the driving mechanism of multi-level indicators of urbanization on the supply and demand of ESs can enhance our understanding of the ecological and environmental impacts of urbanization. This study investigates the driving mechanisms underlying the relationship between urbanization and the supply–demand dynamics of ecosystem services (ESs) in the Yangtze River Middle Reaches Urban Agglomeration (YRMRUA). First, we assessed the variation in the key ESs (food production, carbon storage, and culture service) from 2000 to 2019 at both city and provincial levels. Second, ES demand and the supply–demand index (SDI) were calculated utilizing socioeconomic indicators. The Geographical Detector model was applied to analyze the individual and combined effects of urbanization on the supply and SDI of ESs. The results showed that an increase in areas of supply and demand was unbalanced in the YRMRUA from 2000 to 2019, with a predominant concentration observed in the provincial capital cities. Scale urbanization exhibits the most substantial influence on the SDI, with a $q$-value of 0.6, while land urbanization exerts the most pronounced effect on ES supply, with a $q$-value of 0.7. Furthermore, it is noteworthy that the combined effect of urbanization on ESs surpasses the individual effect, with $q$-values exceeding 0.5. The interaction between scale urbanization and other indicators has the greatest impact on the SDI of carbon storage. Population and economic urbanization exhibit a more substantial impact on food production and cultural service compared to other primary indicators. Simultaneously, the joint effects of secondary indicators between per capita living area and per capita road area have a greater impact on ES supply than other secondary indicators. These findings illustrate that urbanization indicators are not independent of each other, but have a combined effect. Furthermore, the urbanization process in the YRMRUA has exhibited a gradual deceleration, leading to a diminishing influence on ESs. This study can contribute to the comprehension of urbanization and ESs when dealing with the conflict between urban development and ecological sustainability.

**Keywords:** urbanization; ecosystem services; supply and demand; Geographical Detector model

## 1. Introduction

Ecosystem services (ESs) refer to the benefits humans obtain from nature [1,2]. In the past few years, there has emerged an increasing quantity of research focused on the dynamics of ESs and supply–demand relationships globally [3]. Exploring the driving mechanism of ESs and the interrelationship between ESs holds significant relevance in enhancing human well-being [4–7]. An urban agglomeration is characterized as a region

with intensive human activities and a highly developed economy. In the process of urban development, obvious changes occur in the land structure, leading to positive or negative development of ecosystem functions and services [8]. Therefore, it is of great significance to investigate the connection between ecosystem services and urban development [9].

Clarifying the ES supply–demand relationship is paramount in alleviating the contradiction between resources and demand [10]. Incorporating ES supply and demand research into decision making is helpful for effective ecological management [11]. An ES supply–demand mismatch in a region indicates that human demand for ecosystem services cannot be fulfilled to some extent [10]. Mapping the areas experiencing a supply–demand imbalance can help to formulate ecosystem conservation policies [12]. In recent years, extensive research has been conducted on ES supply and demand [13]. Numerous methods exist for evaluating the supply of ESs in terms of monetary and physical quantities, and the methods are relatively mature, including the model method and value equivalent method [14]. Various methodologies are also employed to evaluate demand from a socioeconomic perspective, relying on specific indicators for quantification [15]. Several studies have been carried out to assess the basic ESs in small regions such as cities, river basins, urban agglomerations, or land cover types [16]. Most research has been carried out from the ES supply perspective, with relatively limited exploration from the perspective of human demand. To provide evidence for ecological supply–demand management, assessing ESs in terms of both supply and demand dimensions still needs to be further studied.

The ES supply–demand relationship can be significantly influenced by both natural and anthropogenic factors [17]. It is crucial to gain knowledge of how urbanization influences ESs during the long-term urbanization development process. Due to the increasing demand for the ecological environment, the supply–demand relationship of ecosystems is constantly changing. Urbanization is a process of occupying natural ecosystems, resulting in the degradation and variation in both the structure and function of ecosystems [8]. Furthermore, the urbanization process has continually increased the population and demand for ESs. To make sustainable decisions for the urban ecosystem, the relationship between urbanization and ESs needs to be explored. Recently, a few researchers have studied the urbanization effect on ES supply [18]. Most research focuses on the individual effect of urbanization, with a relative scarcity of comprehensive assessments of urbanization using multi-level indicators. The interactive and complex relationship between urbanization and ESs needs to be further explored in highly developed areas. The methods used to conduct analysis include statistical analysis [19], random forests [20], and the Geographical Detector model. When multiple influencing factors are involved, the Geographical Detector model can effectively tackle the complex computational process without human intervention and is easy to operate [21]. Therefore, the Geographical Detector model was chosen to quantify the relationship between urbanization and ESs in this study.

Various ecosystem services are generated by urban ecosystems, including street trees, lawns/parks, urban forests, cultivated land wetlands, lakes/sea, and streams [22]. Urban ecosystems provide ecosystem services in terms of provisioning, supporting, regulating, and cultural aspects [1]. In this study, considering data availability and prior research, three services were selected from each of the categories of supply, regulation, and cultural services: food production, carbon storage, and cultural service. The Yangtze River Middle Reaches urban agglomeration (YRMRUA) is located in a plain characterized by abundant agricultural land, and the provision of food production services is closely linked to the livelihoods of residents. Assessing the supply and demand of its food production ecosystem services holds significance for enhancing our comprehension of food security. Carbon storage is also another important regulating service, and urban agglomerations typically exhibit higher carbon emissions when compared to the surrounding regions. Trees and other vegetation can sequester substantial amounts of carbon and are often integrated into policies to advance carbon reduction objectives [23]. Furthermore, higher culture services can promote the livability and sustainability of urban centers. Therefore, this study

selects these three ecosystem services for supply and demand assessment and urbanization research.

Over the past few decades, the urbanization process within the YRMRUA has experienced notable acceleration. It is also a prominent economic and transportation hub in central China. To achieve sustainable development goals, revealing the mechanism of urbanization for ES supply and demand at both city and provincial scales and using different urbanization indicators is necessary. The main objectives of this study are (1) to know how the supply and demand of key ESs varied in 2000–2019 in YRMRUA, and (2) to reveal how the urbanization process drives ESs from the perspectives of supply and demand. The results mainly indicate the changing trend of ESs from 2000 to 2019 and reveal the driving mechanism between ESs and urbanization. The research framework and structure are displayed in Figure 1.

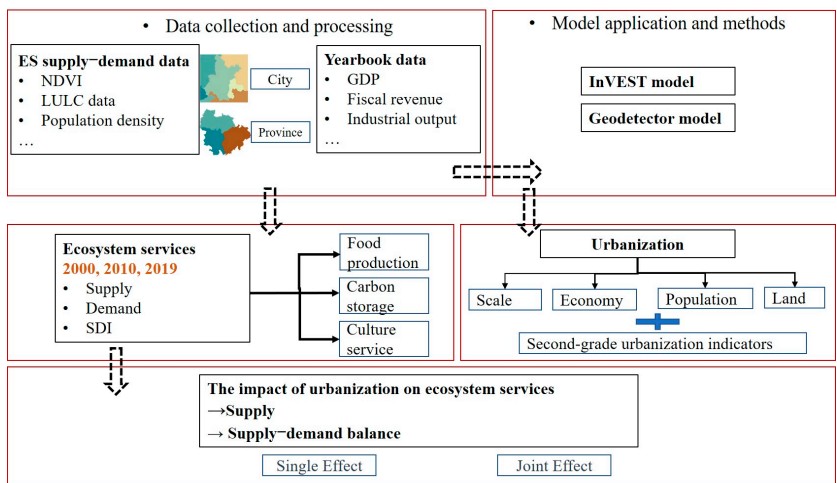

**Figure 1.** The research framework and structure.

## 2. Materials and Methods

### 2.1. Study Area

Covering and area of 326,100 km$^2$, YRMRUA is a national urban agglomeration in central China (Figure 2). The biological diversity of YRMRUA is relatively higher than that of other agglomerations. With the extension of human activities and the acceleration of the urbanization process, this region has undergone dramatic land use/cover changes [24]. A large number of cultivated lands, forest land, and natural types of land have been occupied, and the construction scale has increased steadily. YRMRUA has a subtropical monsoon climate, which is mild and humid, and is characterized by a subtropical–tropical and subtropical moist broadleaf forest. The ecological and environmental problems of YRMRUA are becoming increasingly prominent, with impacts on water, atmosphere, soil, and ecology. Some problems, such as water pollution, carbon emissions exceeding standards, and land use imbalance, constrain the sustainable development.

### 2.2. ESs Supply and Demand

#### 2.2.1. Food Production (Fo_P)

Food production service is assessed using the Crop Production module of the InVEST model (https://naturalcapitalproject.stanford.edu/invest/, accessed on 15 March 2023). The InVEST crop production model comprises a percentile-based yield model encompassing 175 global crop types, along with a regression-based model that incorporates fertilization rates for 10 specific crops. This study used the percentile-based yield model, producing estimates of 175 crops' yields from existing data, percentile summaries, and observed yields. These observations are based on the Food and Agriculture Organization of the United Nations and sub-national datasets for 175 crops, with units of tons/ha. Land use/cover data in 2000, 2010, and 2018 (2019 data are replaced with 2018 data, and all other data are

based on 2019 data) were used in this model, and obtained from Resource and Environment Science and Data Center (http://www.geodata.cn/, accessed on 20 February 2023).

The food demand is calculated by the two indicators of per capita food demand (kg/person) and population density (person/km$^2$), and per capita food demand data (grain and economic crops) are obtained by querying the yearbook data of each city [9].

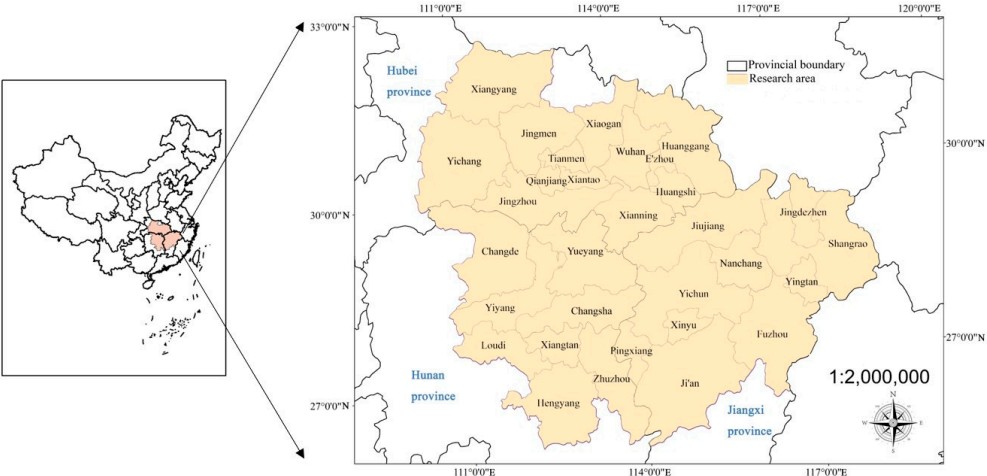

**Figure 2.** The geographical location of YRMRUA.

### 2.2.2. Carbon Storage (Ca_S)

The Carbon Storage module of the InVEST model is used to evaluate carbon storage service (https://naturalcapitalproject.stanford.edu/invest/, accessed on 20 March 2023). Carbon pool data were obtained from the official website of the InVEST model. This module obtains the carbon storage of each grid by summing up the total carbon stored in the carbon pool (including aboveground, underground, soil, and dead organic matter).

Carbon storage service demand was calculated according to carbon emissions, which were estimated using the consumption of standard coal and the carbon emission coefficient of standard coal. The formula used to assess carbon demand is as follows [25]:

$$D_c = (E_a + E_I + E_H) \times C_{coe} \tag{1}$$

where $D_c$ is carbon emissions (ton). $E_a$, $E_I$, and $E_H$ are agricultural, industrial, and household standard coal consumption (t), respectively, obtained from the statistical yearbook. $C_{coe}$ is the carbon emission coefficient of standard coal set by the National Development and Reform Commission, which is 0.68 [9].

### 2.2.3. Culture Service (Cu_S)

The supply of culture service is mainly provided by the blue–green infrastructure area in YRMRUA. The blue–green infrastructure includes forest land, grassland, and water bodies [9,26,27]. Therefore, the proportion of blue–green infrastructure area in urban areas is taken as the quantitative indicator of cultural service supply:

$$S_c = \left( Area_g + Area_f + Area_w \right) / Area_i \tag{2}$$

where $S_c$ is the supply of cultural service, $Area_g$ is the area of grassland, $Area_f$ is the area of forests, $Area_w$ is the area of water bodies, $Area_i$ is the total area of $city_i$.

The demand for cultural services is related to population density and per capita green space area, obtained from the statistical yearbook in each province. Referring to Zhang et al. [9], the cultural service demand is calculated using Formula (3):

$$D_c = \rho_{pop} \times Area_P \tag{3}$$

where $D_c$ refers to the cultural services demand, $\rho_{pop}$ is population density, $Area_P$ refers to the per capita green area.

### 2.2.4. Total Ecosystem Service (TES) and Supply–Demand Index (SDI)

Because the units of the three ESs are different, we standardized the three ESs (Fo_P, Ca_S, and Cu_S) with uniform values between 0 and 1 and added them together to obtain the total ecosystem service [21].

$$TES = \sum_{i=1}^{3} \frac{ES_{i,j} - ES_{i,min}}{ES_{i,max} - ES_{i,min}} \tag{4}$$

where $ES_{i,j}$ is the original value of ecosystem service $i$ in pixel $j$. $ES_{i,min}$ and $ES_{i,max}$ are the minimum and maximum values of ecosystem service $i$, respectively.

The supply–demand index integrates the supply of ESs to real human needs and can reveal surpluses or shortages of ESs, as presented by Li et al. [28]. The formula is as follows:

$$\text{SDI}_i = \frac{S_i - D_i}{(S_{max} + D_{max})/2} \tag{5}$$

where $\text{SDI}_i$ is the supply and demand index for city $i$; $S_i$ and $D_i$ are the ES supply and demand for city $i$, calculated by averaging standardized grid values for each city. $S_{max}$ and $D_{max}$ are the maximum ES supply and demand pixel values of a city, respectively.

### 2.3. Geographical Detector Model

When identifying interactions between driving factors and dependent variables, the Geographical Detector model is an effective tool to measure the driving force in the spatial differentiation mechanism [29]. The Geographical Detector model is widely used because of its advantages of simple operation and uncomplicated data requirements. The factor detector module and interaction detector module in the Geographical Detector model measure the contribution of a single factor to the ESs and the joint effect of paired influencing factors on each ecosystem service, respectively. The $q$-value is used to quantify the extent to which factor X explains the spatial variation of attribute Y. The formula is as follows:

$$q = 1 - \frac{\sum_{h=1}^{L} N_h \sigma_h^2}{N \sigma^2} = 1 - \frac{SSW}{SST} \tag{6}$$

where $h = 1, \ldots, L$ represents the stratification of variable Y or factor X. $N_h$ and $N$ represent the number of units in stratum $h$ and the entire region, respectively. $\sigma_h^2$ and $\sigma^2$ represent the variances of Y values in stratum h and the entire region, respectively. $SSW$ and $SST$ represent the sum of within-group variances (Within Sum of Squares) and the total regional variance (Total Sum of Squares), respectively. The $q$-value ranges from 0 to 1. A higher $q$-value indicates a greater spatial variation in Y. If the stratification is generated by the independent variable X, a larger $q$-value signifies a stronger explanatory power of variable X on attribute Y, while conversely, a smaller $q$-value suggests a weaker explanatory power. The potential limitation of this model is that the Geographical Detector model is unable to determine positive or negative correlation.

In the Geographical Detector model, continuous variables need to be converted into category variables through spatial data discretization. It is worth noting that the data discretization method used in this study is the Geographic Detector 'GD' R package, and the optidis function is used to select the optimal combination of discretization method and quantity by comparing a series of alternative discretization methods and quantities [30].

The urbanization indicators include the primary urbanization indicators and the secondary urbanization indicators, as presented by Zhou et al. [31]. The primary indicators include scale urbanization, economic urbanization, population urbanization, and land use urbanization (Table 1), obtained from the statistical yearbook of three provinces (https://tjj.hubei.gov.cn/, http://tjj.jiangxi.gov.cn/, http://tjj.hunan.gov.cn/ (accessed on 20

April 2023)). This study explored the individual effect and joint effects of urbanization on ES supply and SDI.

**Table 1.** Selection of the primary and secondary indicators of urbanization.

| Primary Indicator | Secondary Indicator | Unit | Abbreviation |
|---|---|---|---|
| Scale urbanization | Total population | $10^4$ person | Pop |
| | GDP | Million CNY | GDP |
| | Built up area | $km^2$ | Build |
| Economy urbanization | GDP per capita | $10^4$ CNY | GDP_p |
| | Per capita financial income | $10^4$ CNY | Inc_p |
| | Per capita industrial output value | $10^4$ CNY | Ind_p |
| | Proportion of tertiary industry | % | Ter |
| Population urbanization | Population density in municipal districts | Person/$km^2$ | Pop_d |
| | Proportion of urban population | % | Pop_u |
| | Per capita expenditure on education | RMB | Edu_p |
| Land urbanization | Per capita living area | $m^2$ | Live_p |
| | Per capita road area | $m^2$ | Road_p |

*2.4. Moran's I Index and Lisa Cluster*

Moran's *I* index in Geoda software ranges from −1 to 1 [32]. The spatial correlation results of urbanization and the ESs can be classified into four types with the Lisa statistical method: high–high (H–H), high–low (H–L), low–high (L–H), and low–low (L–L). H–H, L–L, L–H, and H–L are respectively set as ecological utilization areas (maintain the high coordination between the supply and demand of existing urbanization and ESs, and rationally use ecological resources), ecological reconstruction areas (rationally plan the coordinated development of the future social economy and ecological protection based on sustainable development), ecological source areas (with superior ecological basic resources, strictly protected as ecological source areas), and ecological restoration areas (should improve green infrastructure, strengthen ecological restoration, and improve ecological quality).

**3. Results**

*3.1. Variations in ES Supply–Demand at Different Scales*

From 2000 to 2019, the supply of food production changed slightly at the city scale (Figure 3), ranging from 0 to 0.04 (standardized values). The high values are located in most cities of Hubei and Hunan province. The food demand increased notably, and the high-demand areas expanded from the northern cities in 2000 to most cities in 2019. The food production service of Wuhan is a typical supply–demand imbalance region, but improved in 2019, with SDI increasing from −1.4 to −0.1. However, Jiujiang, Yichun, Shangrao, and Fuzhou changed from supply–demand balance to imbalanced regions. High-value supply areas of carbon storage are distributed in Jiangxi province and most areas of Hunan province, ranging from 0.03 to 0.04, while the areas around Wuhan metropolitan area have a low value (0.02). There are carbon emissions areas with remarkably high values, notably Wuhan (1.15%), Yueyang (1.64%), Jiujiang (4.6%), Nanchang (6.5%), and Yichun (2.5%), while carbon emissions in Jingmen (0.14%), Hengyang (−0.35%), and Loudi (0.02%) decreased or changed slightly. The supply and demand of carbon storage services in most cities of Hubei province, Hengyang, and Loudi city changed from imbalance to balance (from −0.34 to 0.17), while those of Wuhan, Yueyang, and Nanchang changed from balance

to imbalance (from 0.33 to −1.4). The supply of cultural services in most cities in Jiangxi province increased notably, among which Pingxiang (2.0%), Ji'an (1.67%), and Jingdezhen (1.89%) increased the most. The demand for cultural services in Wuhan, Nanchang, and Hengyang increased noticeably, while that in Jiujiang and Xiangyang decreased noticeably. The supply and demand imbalances of cultural service in Wuhan (−0.82), Changsha (−0.18), Xiangtan (−0.2), Hengyang (−0.72), and Nanchang (−0.98) are more prominent, while SDI in Xiangyang and Yichang shifted from imbalance to balance.

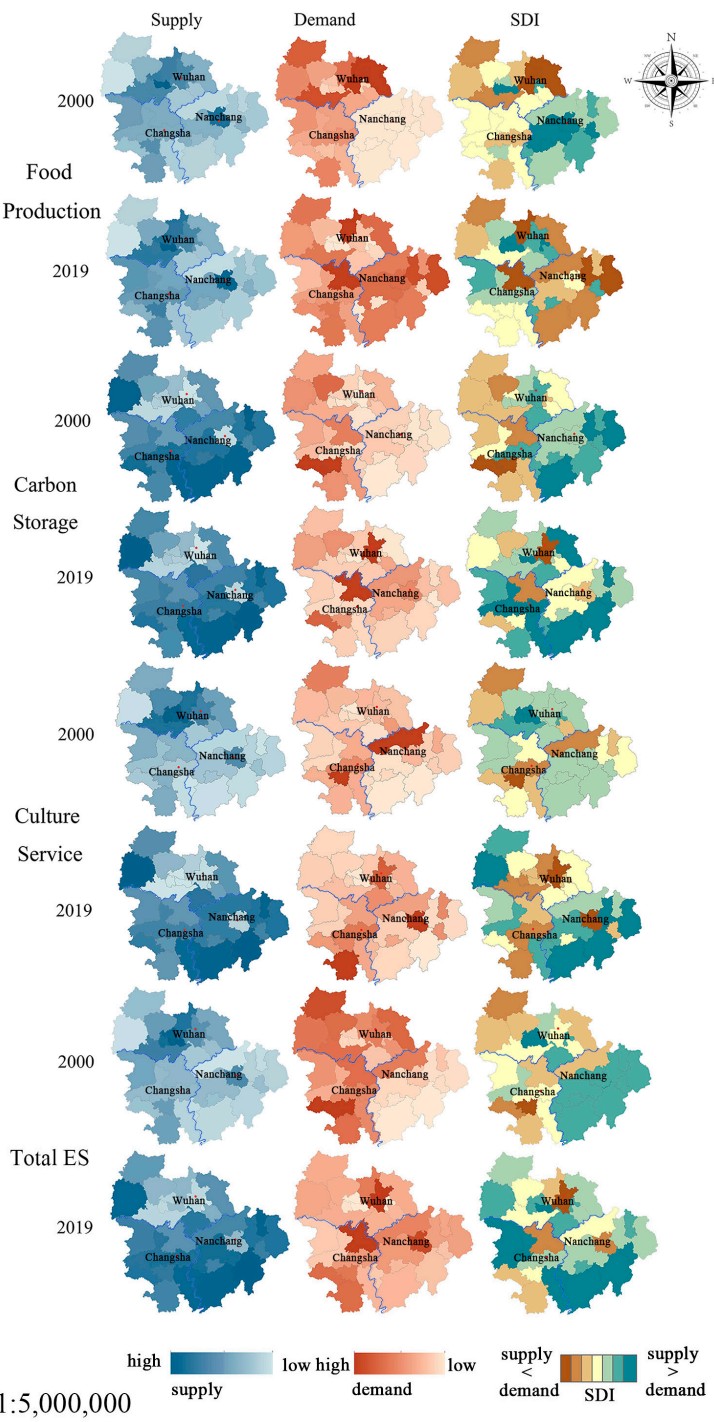

**Figure 3.** Supply, demand, and SDI of ESs at the city scale in YRMRUA in 2000 and 2019.

The provincial scale result (Figure 4) shows that Jiangxi province had a higher demand for food production services in 2019, but the food supply capacity is lower than that of

Hubei and Hunan provinces. Jiangxi province has the highest supply of carbon storage service, while carbon emission is lower than that of the other two provinces. As for cultural services, the supply of Hubei province reduced, and was lower than that of the other two provinces. In Jiangxi province, there exists a notable imbalance between the supply and demand of food production services. Conversely, the supply and demand for cultural and carbon storage services in this region surpass those observed in the other two provinces, indicating an enhanced capacity for cultural services supply and demand. The supply–demand imbalance of cultural services is noticeable in Hunan province. Although the supply–demand imbalance of carbon storage services in the Hubei and Hunan provinces is noticeable, the supply–demand balance of food production services in these two provinces has increased.

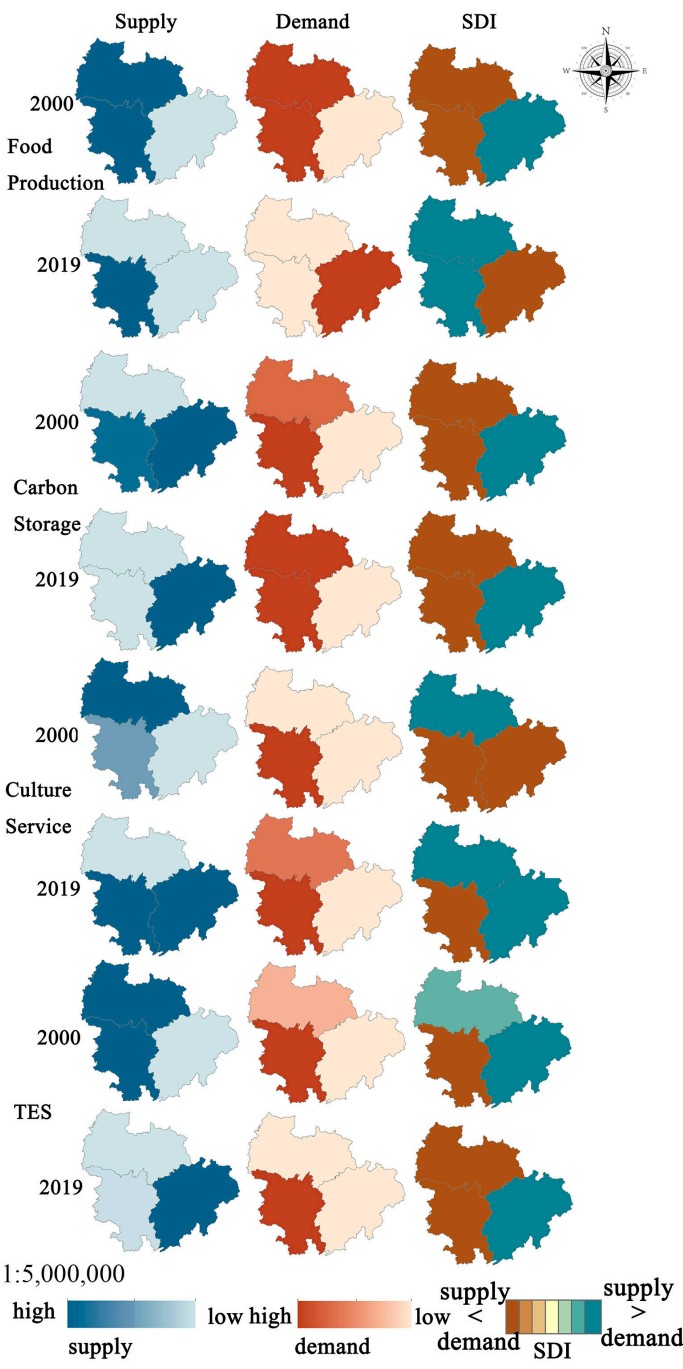

**Figure 4.** Supply, demand, and SDI of ESs at the provincial scale in YRMRUA in 2000 and 2019.

The change in TES (Figure 5) reveals a more pronounced decrease in supply in 2010–2019 than in 2000–2010, and the reduction areas were mainly concentrated in the southern cities. At the provincial scale, the ES supply capacity of Hubei province shows a trend of first decreasing in 2000–2010 and then increasing in 2010–2019, while that of Hunan and Jiangxi provinces shows an opposite trend. The supply and demand balance in Hubei province increased noticeably in 2000–2010 and 2010–2019, while that of Jiangxi province decreased in the two periods.

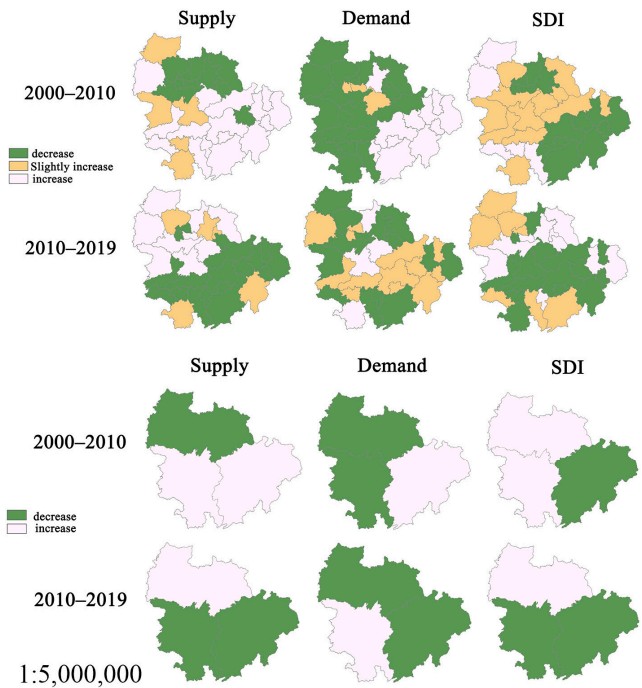

**Figure 5.** Supply, demand, and SDI variations of ecosystem services at the city and provincial scale in YRMRUA in 2000–2010 and 2010–2019.

### 3.2. Simple Effect of Urbanization on ESs

We used the *q*-value to reveal the effect of primary urbanization indicators on the ES supply and demand (Figure 6). Notably, in 2000, the impact of land, population, and economic urbanization on the SDI of total ecosystem services was the most prominent, with a *q*-value of 0.3. By 2019, scale urbanization exerted the most significant effect on the SDI of TES, reaching a *q*-value of 0.6. In 2019, it is noteworthy that scale urbanization exhibited the most substantial impact, with a *q*-value of 0.7, on both food and carbon storage services, surpassing the influence of other indicators. The effect of scale urbanization on culture services is lower than that on other services. Consequently, land, economy, and population urbanization have little effect on ESs, while scale urbanization has the highest effect on SDI, with the largest effect on carbon storage. The impact of secondary urbanization indicators on ESs, as depicted in Figure 7, is lower than that of primary indicators, with an explanatory power of 0.3. The proportion of urban population and total population have the greatest impact on TES but show a declining trend, while the effect of per capita GDP and the proportion of tertiary industry is on the rise. SDI of food production is greatly influenced by per capita GDP and per capita expenditure on education and shows an increasing trend. The proportion of the tertiary industry, per capita education expenditure, and per capita road area significantly influence the SDI of culture services. The SDI of carbon storage is greatly affected by the proportion of urban population and shows an increasing trend from 2000 to 2019; it is also greatly affected by the total population but exhibits a decreasing trend over time. Overall, ESs are affected by different secondary urbanization indicators, but the effect of secondary indicators is lower than that of primary indicators.

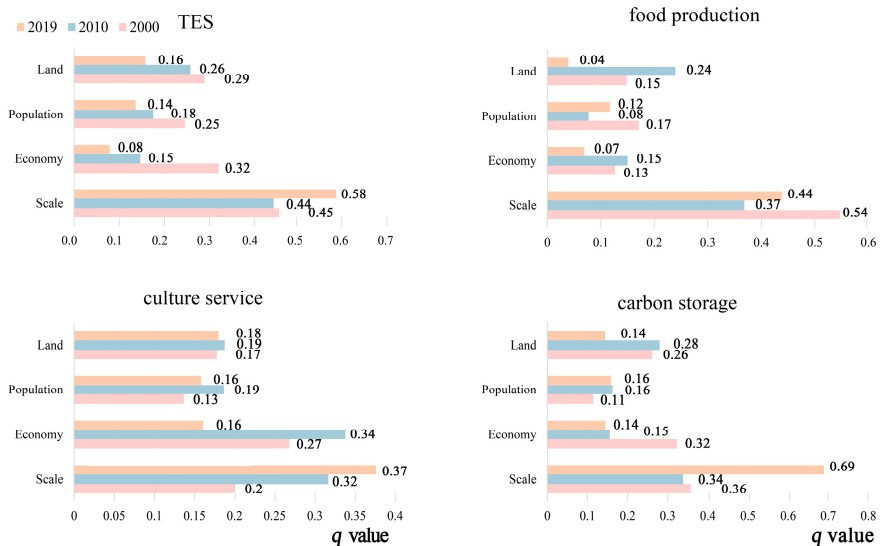

**Figure 6.** *q*-value of primary urbanization indicators to SDI of ESs in 2000, 2010, and 2019.

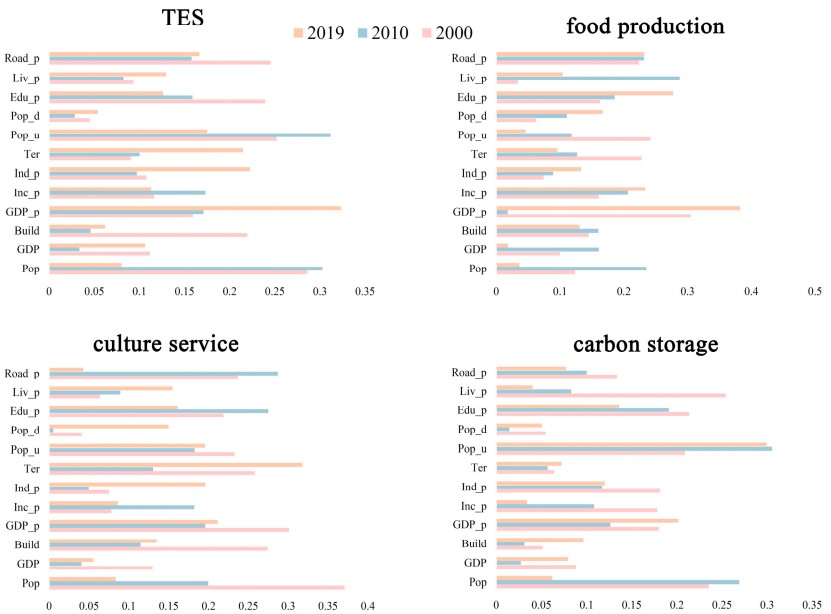

**Figure 7.** *q*-value of secondary urbanization indicators to SDI of ecosystem services in 2000, 2010 and 2019.

Table 2 presents the isolated effects of primary urbanization indicators on ES supply. Specifically, land urbanization exhibited the greatest impact on supply in 2000 with an explanatory power of 0.66. In 2010, economic urbanization emerged as the dominant factor, with the highest explanatory power of 0.46, while in 2019, population urbanization exerted the greatest influence, with an explanatory power of 0.41. In the year 2000, food production services were most significantly influenced by land urbanization, whereas by 2019, population urbanization emerged as the predominant indicator, attaining an explanatory power of 0.41. In 2000, carbon storage services were notably affected by scale, economic, and land urbanization, whereas in 2019 they were greatly affected by land urbanization. Cultural services were greatly affected by population urbanization in 2019 and reached 0.45. Among the four indicators, the three indicators other than scale urbanization have a relatively high impact on the supply of ESs. The individual effect of the secondary indicators of urbanization on the supply of ESs (Table A1) indicates that the effect of population indicators on TES and carbon storage service is relatively high

($q$ > 0.4). The effect of built-up areas on food production and carbon storage service is relatively high. The impacts of per capita living area ($q$ > 0.4), per capita road area ($q$ > 0.5), per capita education expenditure ($q$ > 0.3), and urban population density ($q$ > 0.5) are higher than those of other indicators in 2000 and 2010 but show a weakening trend. Previous research indicated that disparities in educational resources and educational attainment between urban and rural areas result in significant distinctions in ecological knowledge and cognition between urban and rural residents. The overall ecological cognition of urban areas is higher, so more attention is needed to be paid to the protection of ecological land in the process of development [33]. Therefore, the impact of per capita education expenditure on ESs is higher than that of other secondary indicators.

**Table 2.** $q$-value of primary urbanization indicators to the supply of ESs. Bold numbers indicate a higher effect.

| ESs | Year | Scale Urbanization | Economy Urbanization | Population Urbanization | Land Urbanization |
|---|---|---|---|---|---|
| TES | 2000 | 0.32 | 0.36 | 0.11 | **0.66** |
| | 2010 | 0.26 | **0.46** | 0.37 | 0.22 |
| | 2019 | 0.32 | 0.38 | **0.41** | 0.32 |
| Food production | 2000 | 0.36 | **0.40** | 0.15 | **0.67** |
| | 2010 | 0.26 | 0.30 | 0.30 | 0.17 |
| | 2019 | 0.31 | 0.26 | **0.41** | 0.31 |
| Carbon storage | 2000 | **0.46** | **0.57** | 0.22 | **0.70** |
| | 2010 | 0.27 | **0.45** | **0.42** | 0.20 |
| | 2019 | 0.38 | 0.36 | 0.37 | **0.41** |
| Cultural service | 2000 | 0.32 | **0.44** | 0.16 | **0.68** |
| | 2010 | 0.21 | **0.41** | 0.36 | 0.17 |
| | 2019 | 0.32 | 0.30 | **0.45** | 0.31 |

### 3.3. The Joint Effect of Urbanization on ESs

Figure 8 illustrates that the combined impact of primary urbanization indicators on the SDI generally exceeds individual effects, consistently surpassing 0.8. The joint effect of scale urbanization and other primary indicators has a noticeable impact on TES and carbon storage. In 2000, food production and cultural services reflected the combined influence of economic and scale urbanization, while in 2019, these services were primarily affected by the joint effects of land and population urbanization. In addition, a discernible trend of the joint effect occurred over time. Notably, the most pronounced joint effect on SDI in 2000 was attributed to the interplay of scale urbanization and economic and land urbanization. In 2010, land urbanization and economic urbanization had noticeable influences, particularly on cultural services and food production, whereas carbon storage was mainly affected by the combined effects of scale urbanization and other indicators. Nevertheless, in 2019, the combined effect weakened, and only the combined effect of population and other primary indicators retained a noticeable influence on ESs.

The joint effect of urbanization indicators on ES supply is similarly more notable than the individual effect (Figure 9). In 2000, land urbanization, scale urbanization, and economic urbanization yielded the most substantial combined effect on ESs, surpassing 0.9. In 2010, the joint effect of scale urbanization and economic urbanization emerged as the most pronounced indicator, surpassing the impact in 2000, while the joint effect of scale urbanization and the other two indicators exhibited a diminished trend. The isolated effect of scale urbanization is less pronounced compared to its combined effects with other indicators. Moreover, while scale urbanization exhibited a substantial influence in 2000 and 2010, its impact was not notably pronounced in 2019. In 2019, the joint effect of population urbanization, land urbanization, and economic urbanization became particularly noticeable. This further indicates that the urbanization process of YRMRUA

has passed the primary stage characterized by high population concentration and land scale expansion, and entered the middle stage characterized by less population concentration and economic development.

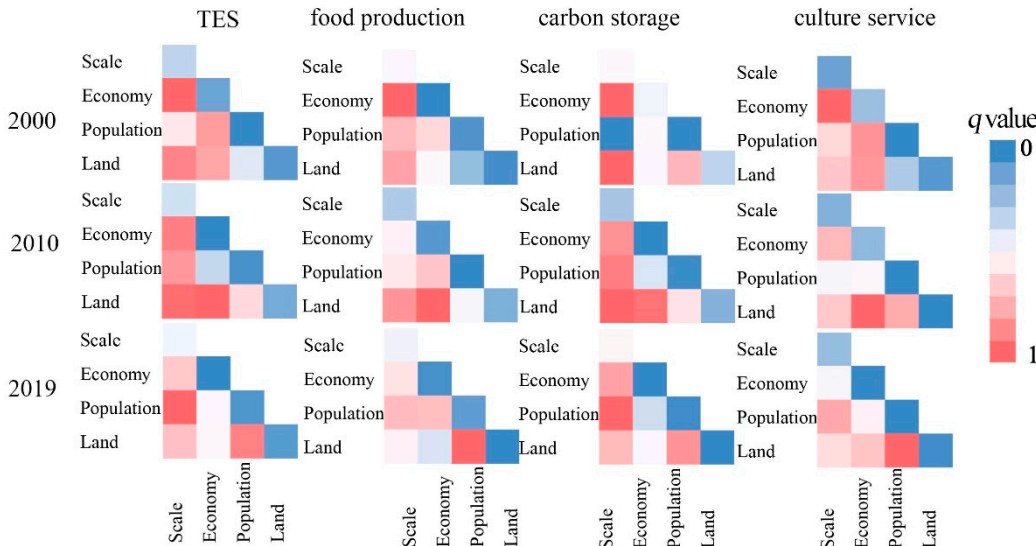

**Figure 8.** The joint effect of primary urbanization indicators on SDI in 2000, 2010, and 2019.

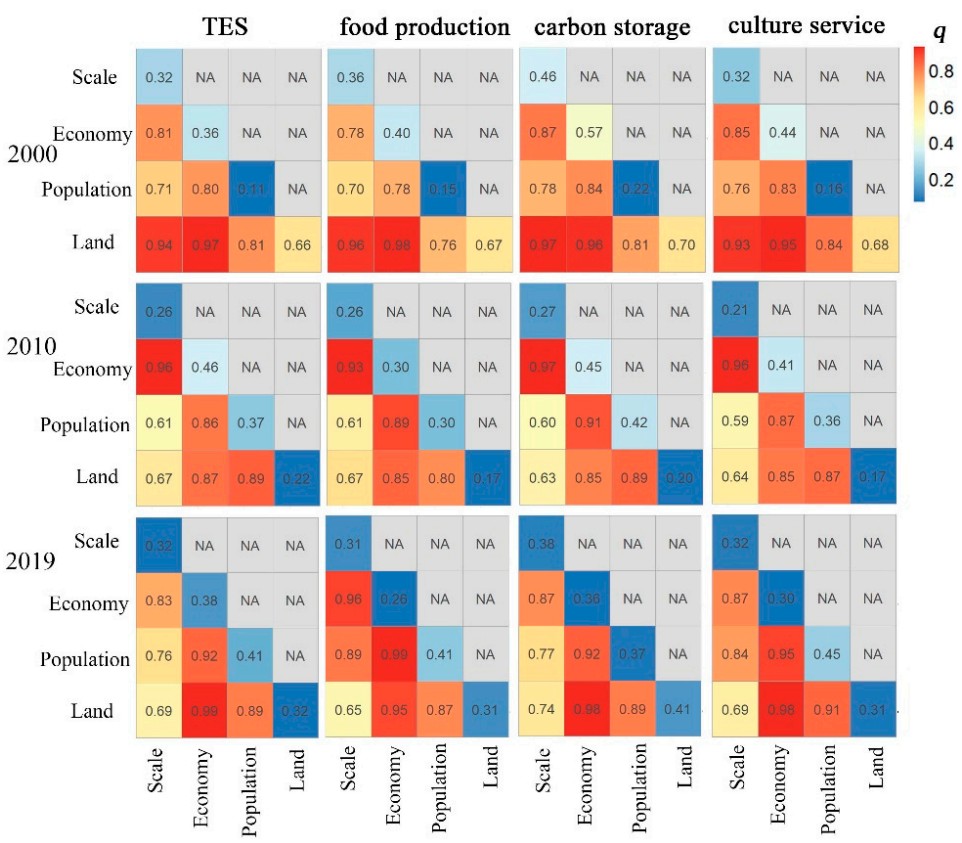

**Figure 9.** The joint effect of primary urbanization indicators on ES supply in 2000, 2010, and 2019.

The combined effect of the secondary indicators of economic urbanization (Figure 10) such as per capita GDP, per capita fiscal income, and scale urbanization in 2000 was substantial but exhibited a weakening trend by 2010. The interaction between the secondary indicators of land urbanization and other secondary indicators notably intensified in 2010 compared to 2000. In 2019, secondary indicators of population urbanization such as the

population density of municipal districts and urban population density, and economic urbanization such as per capita GDP, had a notable combined effect. Overall, the individual effect of the secondary indexes on ESs is modest, while the combined effect of the secondary indexes is notable. In addition, the joint effect exhibited temporal variations. In 2000, it was mainly influenced by the joint effect of the secondary indexes of scale and economic urbanization, while in 2019, it was predominantly influenced by the joint effect of population and economic urbanization.

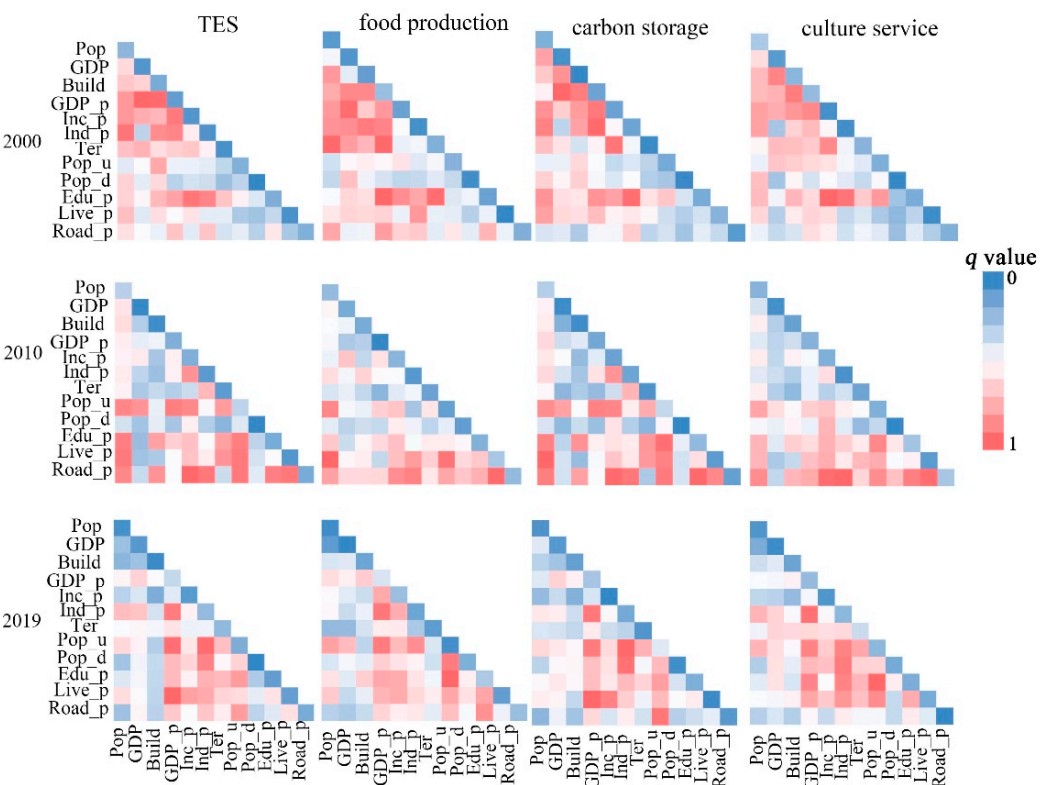

**Figure 10.** The joint effect of secondary urbanization indicators on SDI in 2000, 2010, and 2019.

The joint effect of the secondary indicators of urbanization on ES supply (Figure 11) indicates that the combined effect of the indicators on ESs was most pronounced in 2000, and gradually diminished in 2010 and 2019. Specifically, the combined effects of secondary indicators of land use urbanization, such as per capita road area and per capita living area, and other indicators demonstrated a substantial explanatory power for ES supply in both 2000 and 2010. Furthermore, a notable joint effect was observed between the proportion of tertiary industry and population because the development of tertiary industry mainly occurs on non-ecological land, which is usually in areas with large and concentrated populations. Additionally, the development of the tertiary industry relies heavily on manpower and technology input, and industrial development has a high demand for construction land. In 2010, the combined effect of per capita living area, per capita road area, and other indicators remained robust, and the combined effect of per capita resident income and total population gained prominence compared to 2000. However, in 2019, the combined effect of all indicators weakened relative to the previous years, with notable impacts observed from population density, per capita education funding, and per capita living area of the municipal district on ESs.

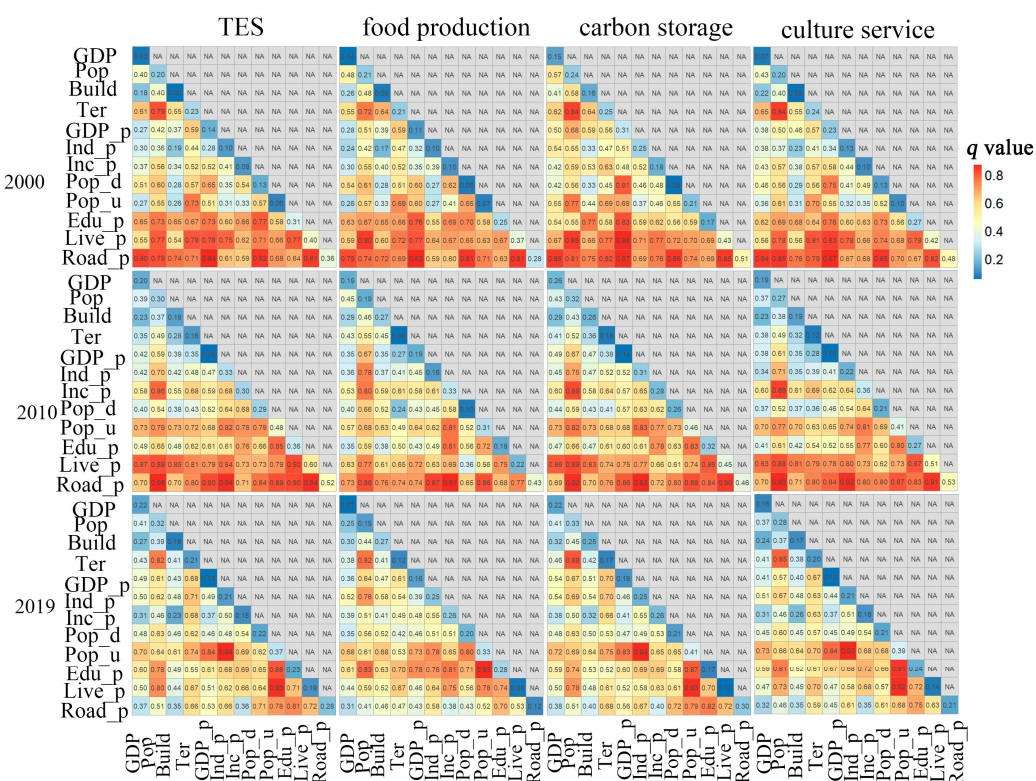

**Figure 11.** The joint effect of secondary urbanization indicators on ES supply in 2000, 2010, and 2019.

### 3.4. Spatial Visualization of Urbanization on SDI

As shown in Figures 12, A1 and A2, scale urbanization and economic urbanization have a negative correlation with SDI (0.37 and −0.24, respectively). This indicates that the high development of scale urbanization and economic urbanization will result in the ES supply–demand imbalance. In 2019, the areas with a concentration of high population urbanization and low SDI were Xiaogan, Jingzhou, Yueyang, Hengyang, and Loudi. By comparison, Jiujiang, Nanchang, Loudi, Hengyang, and Xiangtan are the areas with high land urbanization and low SDI concentration, and also have higher per capita living area and road area but low ES supply and demand. In the Lisa cluster map of scale urbanization and SDI, only several cities in Hubei province had high−high clustering, while most areas had low−high and high−low spatial relationships. The areas with high−low clusters were mainly concentrated in Wuhan, Nanchang, and Yueyang, indicating that these cities had higher urbanization levels but low SDI, thus requiring an optimization of the supply and demand relationship. There are many low−high and low−low cluster cities of economic urbanization and SDI clustering, and a relatively slow economic development speed occurred in most regions of Jiangxi province and Changde and Yiyang cities, which is conducive to achieving supply and demand balance. The areas with high−low economic urbanization and SDI concentration, including Wuhan, Nanchang, Xiangtan, and Jingmen, need to improve their supply–demand balance to adapt to the rapid economic development. In general, it is necessary to focus on the high urbanization but low SDI clustering areas, including Wuhan, Yueyang, Nanchang, Jingmen, Xiangtan, Jingzhou, Xiaogan, Jiujiang, Hengyang, and Loudi. The government needs to strengthen the balance between supply and demand and the optimization of ecological resources in these cities.

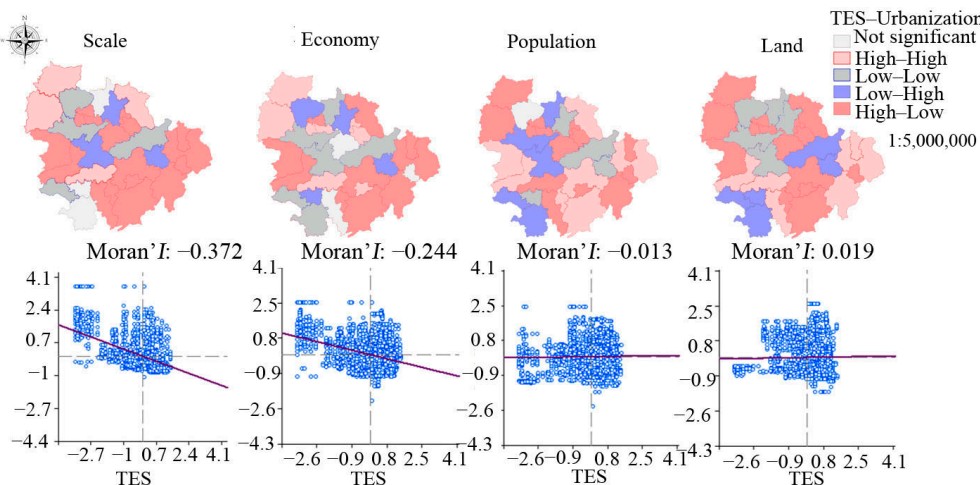

**Figure 12.** Moran's *I* index and Lisa clustering results between urbanization and TES.

## 4. Discussion

### 4.1. Driving Mechanism of Urbanization on Supply–Demand of ES

Over the past two decades, the urbanization process within YRMRUA has exhibited a significant acceleration, leading to noteworthy changes in ESs. Provincial cities witnessed the most extensive urbanization, while other urbanization indicators varied greatly among different cities. The ES variation results can help to identify the hot spots of ES supply to facilitate ecological protection. ESs mainly varied in areas with high population density, followed by increased human demand, remarkable centralization, and a notable supply–demand imbalance in the northern cities. Some studies pointed out that there exists a trade-off relationship between cultivated land and forest land, resulting in the fact that regions with an imbalance in food production supply–demand are regions with balanced culture service supply–demand. The high-value areas of cultural service changed from the Wuhan metropolitan area in 2000 to most cities in Jiangxi province in 2019. The proportion of the blue−green infrastructure area is utilized to measure cultural service, revealing a reduction in the blue−green infrastructure area of the Wuhan metropolitan area in the past 20 years, as confirmed by Wu et al. [34]. Moreover, the supply and demand imbalanced areas of food production and cultural services increased remarkably. In terms of provincial scale, there exist differences in ESs among provinces, and the demand for ESs in Hunan and Hubei provinces is higher than that in Jiangxi province. The supply capacity of Hubei province declined, while that of Hunan and Jiangxi provinces was higher and increased. The ES supply–demand unbalance in Hunan and Hubei is noticeable, but the ES supply–demand balance is much more remarkable in Jiangxi. Expanded knowledge and improvement in ESs in YRMRUA is the obvious difference in the supply and demand imbalance at the provincial level.

We explored the relationship between urbanization and ESs at the city scale (Figure 13). The effect of primary urbanization indicators on ESs was a more pronounced influence than that of the secondary indicators. Specifically, scale urbanization had the greatest negative effect on the SDI of ESs and increased in 2000–2019. With the increasing population, GDP, and built-up area in the past two decades, the demand for ESs increased, consequently exacerbating the supply and demand imbalance. Land urbanization has the smallest impact on SDI, but has the greatest impact on ES supply and shows a weakening trend. Land urbanization mainly affects the supply capacity of food production and cultural services. The expansion of living areas and road infrastructure, along with alterations in green vegetation and farmland cover area, affected ecosystem structure and services. The influence of economic urbanization on ESs showed a weakening trend, while the population urbanization on the supply of ES showed a strengthening trend over time. Economic urbanization expanded the range of human activities and produced a strong

disturbance to the natural ecosystem, thus having a greater impact on the supply of ESs. However, as time goes on, human activities expand to a certain range and start to develop slowly, so the disturbance to the ecosystem begins to weaken. The implementation of ecological policies such as ecological restoration played a role in affecting ESs. With the increasing population, the demand for food services and cultural services increases; thus, food production, agricultural science and technology, and cultural service infrastructure are cultivated to meet human needs. Studies have shown that under the 'strategy for the rise of Central China' in the early stage, urban expansion and development patterns were extensive, leading to the loss of natural resources and further affecting the supply of ESs. However, in the later period, the urbanization strategy and intensity gradually weakened, and the urbanization expansion mode shifted to high-quality development, that is, from the expansion of rough construction land to the improvement in the urban inner living environment [35]. Expanded knowledge about the effects of urbanization on ESs in YRMRUA shows that scale urbanization has the greatest negative effect compared to other urbanization indicators.

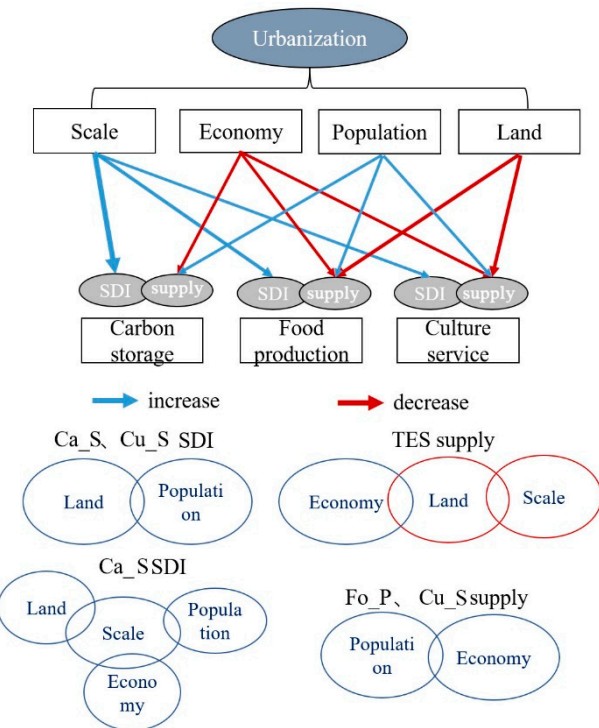

**Figure 13.** The interactive mechanism of ESs and urbanization in the middle reaches of the Yangtze River in 2000–2019.

The combined effect of urbanization indicators is greater than their individual effects, indicating that urbanization indicators are not independent of each other, but jointly act on ESs. Land and population urbanization emerged as major drivers that significantly impact the supply and demand relationship between food production and cultural services. The expansion of urban land is accompanied by population growth, facilitating urban development and subsequently leading to elevated carbon emissions, increased food demand, and cultural service demand [36]. Scale urbanization, along with the other three urbanization indicators, has a great combined effect on the supply–demand relationship of carbon storage. The expansion of the urban scale corresponds to increased economy, population, and land use, leading to an increase in carbon emission, and thereby affecting the supply and demand relationship of carbon storage. The interaction between population and economy has a great influence on food production and cultural supply, and the interaction shows an increasing trend. The interaction between the secondary indicators is also larger than that of the individual factors, and more complex, but the interaction

between SDI and supply certainly shows a remarkable weakening trend. This phenomenon indicates that although the urbanization process in the middle reaches of the Yangtze River has been occurring in the past two decades, it is gradually slowing down, and the interference on ESs demonstrates a pronounced weakening trend over time. Expanded knowledge about the effects of urbanization on SDI in YRMRUA shows that urbanization indicators are not independent of each other, but have a combined effect.

*4.2. Policy Implications and Limitations*

In terms of the degradation of ESs and the supply–demand imbalance areas, the optimization of blue−green space should be considered to meet people's increasing demand for cultural services. The unbalanced areas of food production are mainly concentrated in Jiangxi province. Attention should be paid to the protection of locally cultivated land and food security. The increasing carbon emissions intensified the supply and demand imbalance of carbon storage services, especially in provincial capitals. Therefore, it is necessary to strengthen the construction of carbon sink land in provincial capitals and hinterlands, reduce carbon emissions, and pay attention to the use of clean energy. According to the Lisa clustering results, ecological zoning can be carried out. The high urbanization and low SDI region is set as an ecological restoration area, mainly including Wuhan, Yueyang, Nanchang, Jingmen, Xiangtan, Jingzhou, Xiaogan, Jiujiang, Hengyang, and Loudi. The government needs to strengthen the balance between supply and demand and the optimization of ecological resources in these cities. It should improve the construction of green infrastructure, strengthen ecological restoration, and improve ecological quality. Gomez-Baggethunet et al. [37] found that people in densely populated urban areas are more willing to pay for ecosystem service protection. In this study, scale urbanization includes the total population and has the highest effect on ESs. Therefore, it is advisable to consider implementing appropriate ecological payment policies in densely populated regions to ensure the maximization of urban ecological benefits.

The categories with high urbanization and high SDI were set as ecological utilization areas, including Changsha, Yichang, Xiangyang, Ezhou, Changde, Yiyang, Yichun, Zhuzhou, and Fuzhou. It is suggested to maintain the high coordination between the existing urbanization and the supply and demand of ESs in this kind of area and make rational use of ecological resources. Areas with low urbanization and high SDI are set as ecological source areas, such as most areas in southern Jiangxi province, which have superior ecological basic resources. Therefore, it is recommended to pay attention to the functions of ecological source areas and implement strict protection.

There are some limitations in this study. Given that complete socioeconomic data only are available at the municipal level, this study only considered urbanization and ES supply/supply–demand of a single city. However, at different spatial scales, the relationship may be different. Second, the InVEST model has limitations and cannot verify the simulation results. The method used to calculate the supply–demand balance of carbon storage deviates from the actual result. At present, the prediction and ES analysis of urban agglomeration is a popular research topic. However, this study only focuses on the driving mechanisms of urbanization in the past. Furthermore, the climate factor is also an important factor impacting ESs, but this study lacks relevant discussion about climate elements. There is a certain relationship between carbon emissions and climate change [38]. For example, Elahi et al. [39] studied the relationship between installing renewable energy technology and farmers' willingness, which has certain references for reducing agricultural carbon emissions and climate change. Knight et al. [40] explored the relationship between economic growth and carbon dioxide emissions and found that the effect of economic growth is greater for consumption-based emissions than territorial emissions. Therefore, more research linked to climate change and ecosystem services is needed in the future.

## 5. Conclusions

This study revealed the single and combined influences of urbanization indicators on ESs. Three main conclusions are drawn: (1) The ESs mainly varied in areas with high population density, characterized by increased demand, increasingly noticeable centralization, and notable supply–demand imbalances in the northern cities of the study area. (2) The combined effect of the urbanization indicators is more intense than their individual effect. Notably, land urbanization and population urbanization indicators have a great impact on the supply and demand of food production and cultural services. (3) The supply–demand unbalanced areas of food production are primarily concentrated in Jiangxi province. The high urbanization and low SDI region is set as an ecological restoration area, mainly including Wuhan, Yueyang, and Nanchang. The government needs to pay more attention to these cities and strengthen the balance between supply and demand and the optimization of ecological resources.

**Author Contributions:** J.G.: Writing—Original draft preparation, revision. X.D.: Conceptualization, Methodology, Software. L.W.: Reviewing and Editing, Supervision, Project administration, Funding acquisition, revision. Z.N.: Methodology, Software, Supervision. Q.C.: Methodology, Software. C.H.: Reviewing and Supervision. All authors have read and agreed to the published version of the manuscript.

**Funding:** This work was financially supported by the National Natural Science Foundation of China (41975044, 42371354, 41801021, 42101385), Open Fund of Hubei Luojia Laboratory (2201000043), and the Fundamental Research Funds for National Universities, China University of Geosciences, Wuhan.

**Data Availability Statement:** The land use/cover data was provided by Resource and Environmental Data Center of the Chinese Academy of Sciences (https://www.resdc.cn/, accessed on 20 February 2023). The socioeconomic data, including per capita food demand data, standard coal consumption, per capita green area, population density, primary indicators and secondary indicators of urbanization, provided by the Jiangxi, Hubei and Hunan Provincial Bureau of Statistics (https://tjj.hubei.gov.cn/, http://tjj.jiangxi.gov.cn/, http://tjj.hunan.gov.cn/, accessed on 20 April 2023).

**Conflicts of Interest:** The authors declare no conflict of interest.

## Appendix A

**Table A1.** $q$-value of secondary urbanization indicators to the supply of ESs in 2000, 2010 and 2019.

| Primary Indicators | Year | Scale Urbanization | | | | Economy Urbanization | | | Population Urbanization | | | Land Urbanization | |
|---|---|---|---|---|---|---|---|---|---|---|---|---|---|
| | | GDP | Pop | Build | Ter | GDP_p | Ins_p | Inc_p | Pop_d | Pop_u | Edu_p | Liv_p | Road_p |
| TES | 2000 | 0.03 | 0.20 | 0.02 | 0.23 | 0.14 | 0.10 | 0.09 | 0.13 | 0.06 | 0.31 | 0.40 | 0.36 |
| | 2010 | 0.20 | 0.30 | 0.18 | 0.16 | 0.09 | 0.33 | 0.30 | 0.29 | 0.48 | 0.36 | 0.60 | 0.52 |
| | 2019 | 0.22 | 0.32 | 0.16 | 0.21 | 0.13 | 0.21 | 0.19 | 0.22 | 0.37 | 0.23 | 0.19 | 0.28 |
| Food production | 2000 | 0.04 | 0.21 | 0.06 | 0.21 | 0.11 | 0.10 | 0.10 | 0.06 | 0.07 | 0.25 | 0.37 | 0.28 |
| | 2010 | 0.19 | 0.19 | 0.27 | 0.08 | 0.19 | 0.16 | 0.33 | 0.10 | 0.31 | 0.18 | 0.22 | 0.43 |
| | 2019 | 0.07 | 0.15 | 0.27 | 0.12 | 0.16 | 0.25 | 0.26 | 0.20 | 0.33 | 0.28 | 0.08 | 0.12 |
| Carbon storage | 2000 | 0.15 | 0.24 | 0.16 | 0.25 | 0.31 | 0.25 | 0.18 | 0.06 | 0.21 | 0.17 | 0.43 | 0.51 |
| | 2010 | 0.26 | 0.32 | 0.26 | 0.16 | 0.14 | 0.31 | 0.28 | 0.26 | 0.46 | 0.32 | 0.45 | 0.46 |
| | 2019 | 0.22 | 0.34 | 0.25 | 0.17 | 0.19 | 0.25 | 0.26 | 0.21 | 0.41 | 0.17 | 0.13 | 0.30 |
| Culture service | 2000 | 0.07 | 0.20 | 0.05 | 0.24 | 0.23 | 0.13 | 0.10 | 0.13 | 0.10 | 0.27 | 0.42 | 0.48 |
| | 2010 | 0.19 | 0.27 | 0.19 | 0.12 | 0.10 | 0.22 | 0.36 | 0.22 | 0.41 | 0.28 | 0.51 | 0.53 |
| | 2019 | 0.16 | 0.28 | 0.17 | 0.20 | 0.12 | 0.21 | 0.18 | 0.21 | 0.39 | 0.24 | 0.14 | 0.21 |

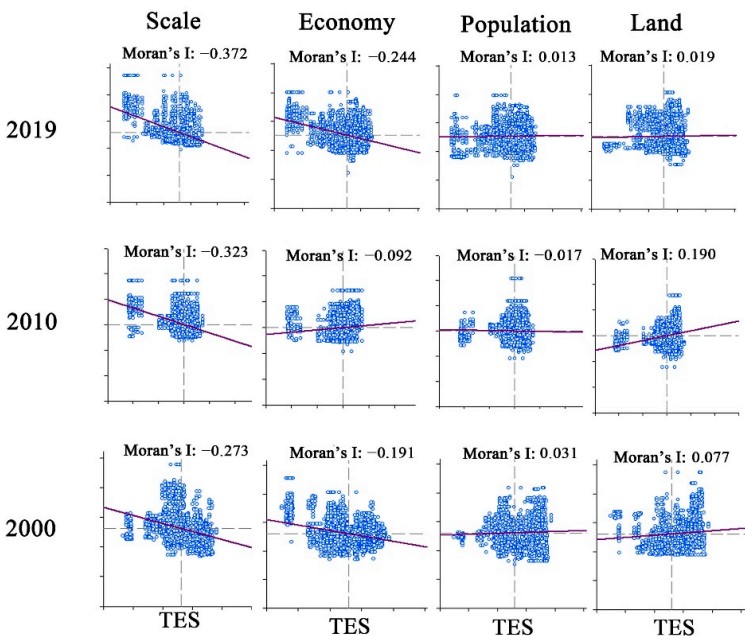

**Figure A1.** Moran's *I* index between different urbanization indicators and TES in 2000, 2010, and 2019.

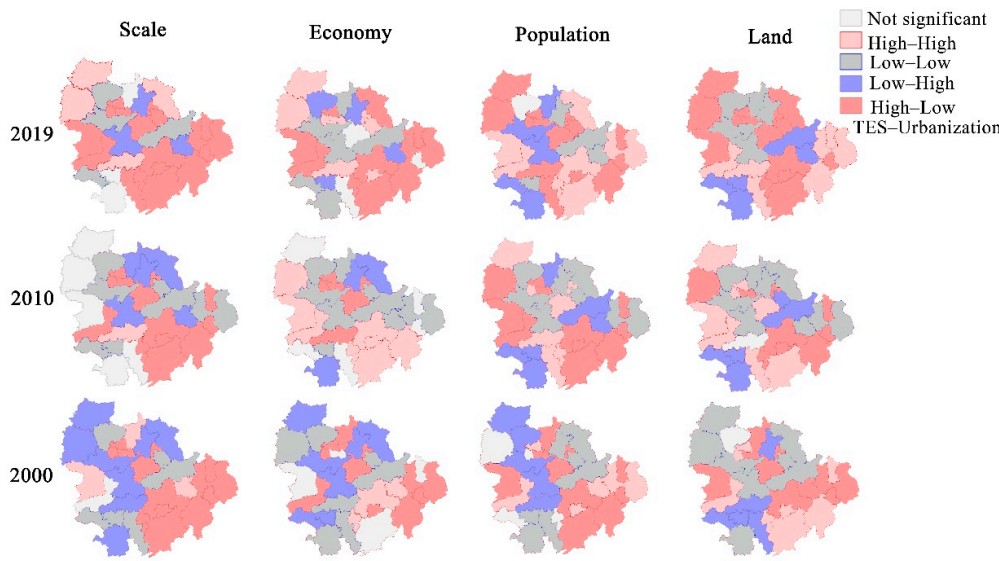

**Figure A2.** Lisa clustering results for 2000, 2010, and 2019.

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
