# Peer review of "The Impact of Urbanization on the Supply–Demand Relationship of Ecosystem Services in the Yangtze River Middle Reaches Urban Agglomeration"

_remotesensing, doi:10.3390/rs15194749_

Round 1

Reviewer 1 Report

The provided abstract outlines a study focused on understanding the impact of urbanization on the supply-demand relationship of ecosystem services (ESs) in the Yangtze River Middle Reaches urban agglomeration (YRMRUA). Here are some critical comments on the article:

Clarity and Organization: The abstract is clear and well-organized, effectively summarizing the key components of the research, including the objectives, methods, and main findings. However, it could benefit from a more concise description of the research methodology to maintain brevity.

Research Significance: The study addresses an important and timely topic – the ecological and environmental effects of urbanization. Urbanization is a global phenomenon with significant implications for sustainability, and understanding its impact on ecosystem services is crucial.

Data and Analysis: The use of diverse indicators and spatial scales in the analysis demonstrates a comprehensive approach to studying urbanization's impact on ESs. However, it would be beneficial to include a brief explanation of the data sources and methods used for data collection and analysis to provide transparency and help readers assess the reliability of the findings.

Quantitative Analysis: The application of the Geographical Detector model is mentioned, which is a robust method for assessing spatial relationships. It would be helpful to provide more information on how this model was applied and any potential limitations associated with its use.

Results and Implications: The results mention key findings, such as the increasing supply-demand imbalance in certain areas and the influence of different facets of urbanization on ESs. However, it would be more informative if some specific quantitative results or effect sizes were included to provide a sense of the magnitude of these effects.

Policy and Practical Implications: While the study concludes that urbanization has an impact on ESs, it would be valuable to briefly discuss the policy and practical implications of these findings. How can this research inform urban planning or environmental management in the YRMRUA or similar regions?

Further Research: The abstract hints at potential interactions between urbanization indicators and ESs but does not delve into their mechanisms or causality. Suggesting areas for further research or elaborating on how the findings can contribute to a deeper understanding of these relationships would be beneficial.

Overall, the abstract provides a solid overview of the research, its objectives, and key findings. Expanding on certain aspects, such as research methods and policy implications, could enhance its clarity and relevance to a broader audience interested in urbanization and ecosystem services.

The last statement of section 4 must have to update with given studies as “There is a certain relationship between carbon emissions and climate change [1,2], and more research linked to climate change is needed in future research.

[1] Understanding farmers’ intention and willingness to install renewable energy technology: A solution to reduce the environmental emissions of agriculture

[2] Economic growth and climate change: a cross-national analysis of territorial and consumption-based carbon emissions in high-income countries

The article is required a proofread to correct its English. 

Author Response

We are very grateful to you for the valuable comments on our revised manuscript. According to your suggestions, we have carefully revised the manuscript in keeping with the comments as much as possible. We have revised every part mentioned by four reviewers. The English grammar and expression in this article have been significantly improved. Altogether, we have improved our limitations, and think this manuscript is better than the last revised one.

Reviewer 2 Report

Dear Authors,

It was a pleasure to review your work. One of its strengths is a clear description of the methodology. I have a question regarding the standardization of units which you mention in line 156. I suggest adding one or two sentences to explain the process.

For better clarity, you might consider placing all Figures in the main body, not Appendices. It was confusing for me to follow the double numbering, scrolling up and down several times.

The article should be proofread to eliminate minor errors, e.g., abbreviations in the last row of Table 1 are mismatched. 

The use of the English language is proper. General proofreading should be sufficient to remove minor errors.

Author Response

(The authors gave the same response as above.)

Reviewer 3 Report

Overall, this paper provides interesting research on the impact of urbanization on ecosystem service supply and demand in the Yangtze River midstream urban agglomeration. However, there are several areas that need improvement and clarification. Below are specific comments: 

Firstly, the English language of this paper needs polishing. Some expressions, such as "first-grade indicators" and "second-grade indicators," do not seem to align with common English usage.

 It is unclear why the author chose to study the period from 2000 to 2019 instead of 2000 to 2020. Additionally, the use of 2019 data seems inconsistent since it appears that 2018 data is also utilized. It would be better to consider studying the changes from 2000 to 2018 directly. 

Section 2.2.1 on "Food production services" could benefit from providing specific formulas, similar to the approach used in sections 2.2.2 and 2.2.3. This would make the analysis more clear and precise. 

The calculation method of the SDI, as described in Formula 4, seems to have a parallel relationship with the indices in sections 2.2.1 to 2.2.3. It would be more appropriate to present the calculation process of the SDI as a separate section, such as 2.2.4.

 The variables in Formula 4 are not fully and clearly explained. For instance, the author mentions that "si" is obtained through the zonal statistics method, but it is not clarified based on what data this method was implemented. The author also mentions "Di" being obtained from statistical yearbooks, but it is questionable whether this indicator is available in such yearbooks. Additionally, how the "ES supply and demand value" of a city is calculated for "Smax" and "Dmax" remains unclear.

 It is unclear whether Formula 4 is the author's own design or derived from previous research. The formulation appears to lack a logical basis.

 The term "scale urbanization" seems uncommon. It is unclear whether the authors have introduced this term or if it has been mentioned in previous research. 

Many indicators listed in Table 1 seem unrelated to urbanization. It is recommended that the authors optimize these indicators to strengthen their association with urbanization. For example, total population, GDP, and built-up area, mentioned under "Scale Urbanization," do not seem highly correlated with urbanization. Considering the research focuses on the Yangtze River midstream urban agglomeration, an important agricultural region in China, rural economy and agricultural production are crucial. Therefore, regions with higher total population, GDP, and built-up area do not necessarily reflect the degree of urbanization in this region. Similarly, GDP per capita and per capita expenditure on education do not have a direct correlation with urbanization.

 In Table 9, the abbreviations for "Per capita living area" and "Per capita road area" under "land urbanization" seem to be reversed.

Moderate editing of English language required

Author Response

(The authors gave the same response as above.)

Reviewer 4 Report

In this article, the driving mechanism between urbanization and ESs supply-demand relationship in the Yangtze River Middle Reaches urban agglomeration was explored. The Geographical Detector model is chosen to quantify the relationship between urbanization and ESs in this study. The paper is well written, but there are some problems that need to be improved. Some specific suggestions and comments are as follows.

1. Introduction: What aspects should ecosystem services be evaluated? The author should add some references.

2. Line 77-82: The methods used to drive analysis should be further supplemented, such as geographical weighted regression, and panel regression analysis. Other methods also have many advantages. Why Geographical Detector model was chosen to quantify the relationship between urbanization and ESs?

3. Section 2.2: Can food production, carbon storage, and culture service represent ecosystem services?

4. Line 109-111: Why 31 cities were divided into three types? Is this division scientific?

5. Line 156-158: Fo_P, Ca_S, and Cu_S were directly added. Do all three indicators have the same weight?

6. Table 1: Since "GDP per capita=GDP/Total population", the GDP per capita may be redundant.

7. Table 1: I think the selection of indicators is not enough, and the relationship between indicators needs in-depth analysis.

8. Figure 3 and Figure 10: Cartographic elements such as compass and scale should be added.

The quality of English language are fine.

Author Response

(The authors gave the same response as above.)

Round 2

Reviewer 1 Report

The authors have addressed all previous comments, and paper is ready to publish in its current form. 

Author Response

Thank you, we revised and checked the reference format again and added the page number and literature number, also updated the doi format.

Reviewer 3 Report

Authors have addressed my concerns.

Minor editing of English language required

Author Response

Thanks for your comment. We checked the English grammar and revised some sentences, and deleted some redundancy. We also ask scholars with good English proficiency to help improve the English language in the paper. I hope the revised version can satisfy the requirements.

Reviewer 4 Report

Overall, the authors have made revisions to the paper according to my suggestions. Some minor issues need to be further addressed.

Some references missing literature number or page number, and the representation of doi is also inconsistent.

The quality of English language is fine.

Author Response

Thank you for your carefulness, we revised and checked the reference format again and added the page number and literature number, also updated the doi format.